# A conserved glycine harboring disease-associated mutations permits NMDA receptor slow deactivation and high Ca$^{2+}$ permeability

Johansen B. Amin[1,2,3], Xiaoling Leng[4], Aaron Gochman[5], Huan-Xiang Zhou[4,6] & Lonnie P. Wollmuth [3,5,7]

A variety of de novo and inherited missense mutations associated with neurological disorders are found in the NMDA receptor M4 transmembrane helices, which are peripheral to the pore domain in eukaryotic ionotropic glutamate receptors. Subsets of these mutations affect receptor gating with dramatic effects, including in one instance halting it, occurring at a conserved glycine near the extracellular end of M4. Functional experiments and molecular dynamic simulations of constructs with and without substitutions at this glycine indicate that it acts as a hinge, permitting the intracellular portion of the ion channel to laterally expand. This expansion stabilizes long-lived open states leading to slow deactivation and high Ca$^{2+}$ permeability. Our studies provide a functional and structural framework for the effect of missense mutations on NMDARs at central synapses and highlight how the M4 segment may represent a pathway for intracellular modulation of NMDA receptor function.

[1] Graduate Program in Cellular and Molecular Pharmacology, Stony Brook University, Stony Brook, NY 11794-5230, USA. [2] Medical Scientist Training Program (MSTP), Stony Brook University, Stony Brook, NY 11794-5230, USA. [3] Center for Nervous System Disorders, Stony Brook University, Stony Brook, NY 11794-5230, USA. [4] Department of Physics and Institute of Molecular Biophysics, Florida State University, Tallahassee, FL 32306, USA. [5] Department of Neurobiology and Behavior, Stony Brook University, Stony Brook, NY 11794-5230, USA. [6] Departments of Chemistry and Physics, University of Illinois at Chicago, Chicago, IL 60607, USA. [7] Department of Biochemistry and Cell Biology, Stony Brook University, Stony Brook, NY 11794-5230, USA. Correspondence and requests for materials should be addressed to L.P.W. (email: lonnie.wollmuth@stonybrook.edu)

Aberrant ion channel functions, or channelopathies, are a major cause of neurological disorders[1]. Ionotropic glutamate receptors (iGluRs), including AMPA (AMPAR) and NMDA (NMDAR) receptor subtypes, are glutamate-gated ion channels found throughout the brain and participate in almost all brain functions. In terms of channelopathies, iGluR autoantibodies are associated with paraneoplastic encephalopathies and the neurological dysfunction in neuropsychiatric lupus[2,3]. Further, numerous de novo and inherited missense mutations have been identified in iGluR subunits that are associated with a wide spectrum of disorders, including epilepsy, intellectual disability, movement disorders, and schizophrenia[4–7].

iGluRs are highly modular proteins composed of four domains (Fig. 1a) and function as tetrameric complexes[8]. Interestingly, missense mutations in the ligand-binding domain (LBD) and transmembrane domain (TMD) forming the ion channel are typically associated with more severe clinical phenotypes[7,9], reflecting their central role in agonist binding and ion channel opening. The ion channel core is formed by two transmembrane

helices, M1 and M3, and an intracellular reentrant pore helix, M2[10,11]. In eukaryotic iGluRs, there is an additional transmembrane helix, M4, that packs against the M1 and M3 helices of a neighboring subunit. The M4 helix plays an important role in receptor biogenesis and ion channel function[12–15].

In the NMDAR M4 segments, there are numerous missense mutations associated with clinical pathologies (Fig. 1b, c; Supplementary Table 1). These mutations are primarily located in the extreme extracellular third of the M4 segments (Fig. 1c), which participates strongly in ion channel gating[16,17], implicating gating deficits as a potential cause for these disease phenotypes.

Here we address the contribution of these M4 missense mutations to receptor function. Although many of these mutations impact receptor gating, the most notable and robust effects occur at a highly conserved glycine (Supplementary Figure 1) positioned near the extracellular end of the M4 segments. Missense mutations at this position in the GluN1 and GluN2B subunits have been identified in multiple patients, and the resulting changes in side chain properties can be either great (glycine-to-

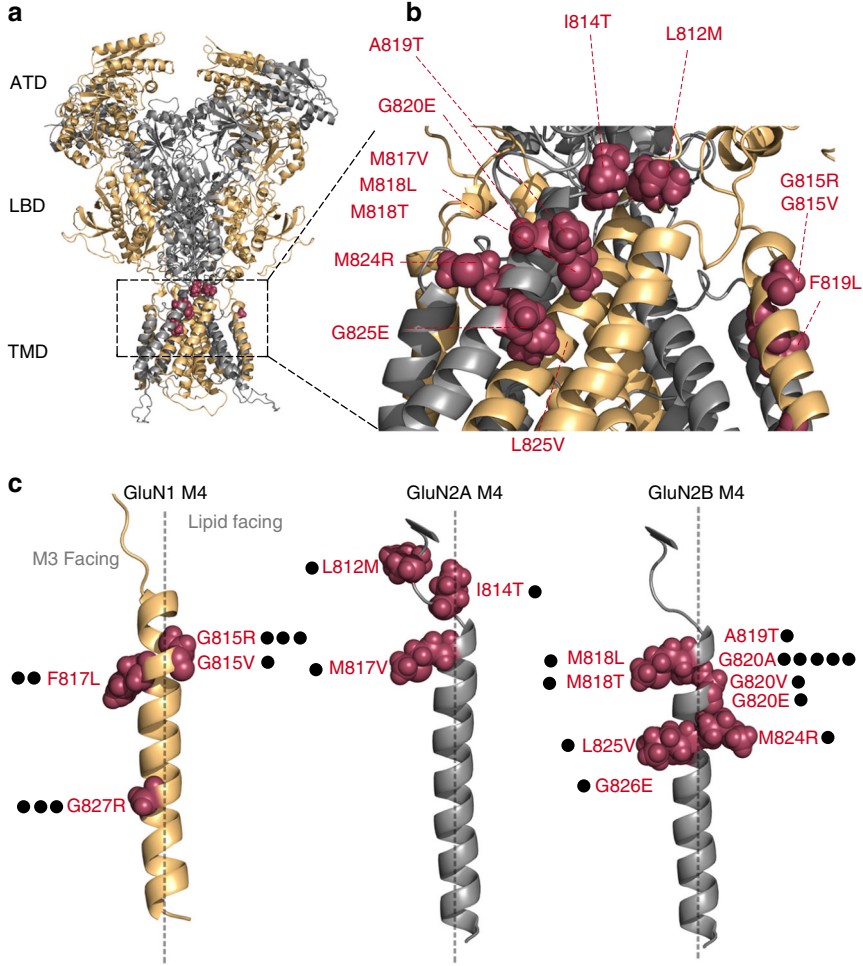

**Fig. 1** Topology of NMDA receptors and distribution of missense mutations in and around the M4 transmembrane segments. **a** Model NMDAR structure (structure based on 4TLM of GluN1/GluN2B)[16] lacking the intracellular C-terminal domain (CTD). Subunits are colored light orange (GluN1) and gray 60% (GluN2). For iGluRs, the tetrameric complex is composed of four highly modular domains: the extracellularly located amino-terminal (ATD) and ligand-binding (LBD) domains; the membrane-spanning transmembrane domain (TMD) forming the ion channel; and the CTD. Positions highlighted in magenta are disease-associated missense mutations in the GluN1, GluN2A, or GluN2B M4 transmembrane segment or the S2-M4 linker (L812M and I814T in GluN2A). **b** An enlarged view of the extracellular end of the TMD highlighting the distribution of missense mutations in and around the M4 segments. **c** The GluN1, GluN2A, and GluN2B M4 segments indicting the specific location of missense mutations. Each dot represents a patient in which the mutation has been identified (Supplementary Table 1). The dashed line separates M3 (leftside) versus lipid (rightside) facing portions of the M4 transmembrane segment

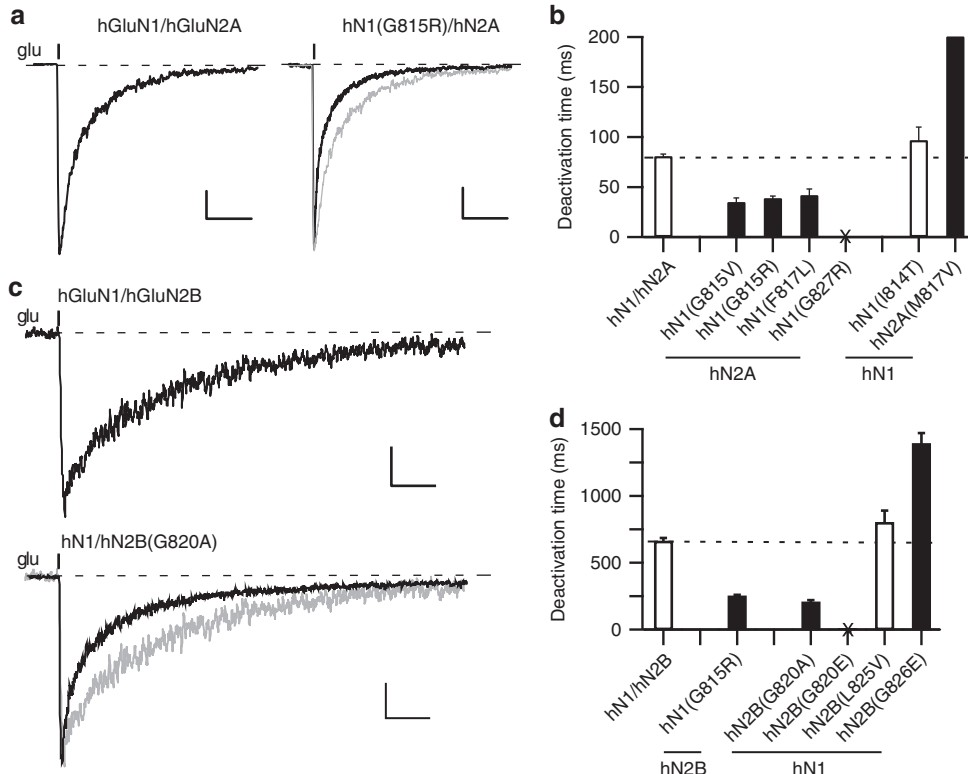

**Fig. 2** Missense mutations in M4 segments alter NMDAR gating. **a**, **c** Whole-cell currents from HEK293 cells expressing human NMDAR subunits, either hGluN1/hGluN2A (**a**) or hGluN1/GluN2B (**c**) or the same receptor with a missense mutation at the conserved glycine. Currents were elicited by a 2 ms application of glutamate (1 mM) in the continuous presence of glycine (0.1 mM), as approximately occurs at synapses. Gray traces are wild type. Holding potential, −70 mV. Scale bars: (**a**) 100 pA, wild type; 25 pA, hN1(G815R); (**c**) 50 pA; time base was 0.2 s for all. **b**, **d** Bar graphs (mean ± SEM) ($n > 5$) showing deactivation rates for hGluN1/hGluN2A (**b**) or hGluN1/hGluN2B (**d**) (Supplementary Table 2). No whole-cell current could be detected for hGluN1 (G827R)/hGluN2A or hGluN1/hGluN2B(G820E), which were tested on at least three different transfection cycles (Supplementary Figure 2). Solid bars indicate values significantly different from wild type ($p < 0.05$, $t$-test)

arginine) or subtle (glycine-to-alanine). For both NMDAR subunits, but most notably the obligatory GuN1 subunit, mutations at this conserved glycine drastically impair the stability of long-lived open states, greatly enhancing the rate of receptor deactivation as well as attenuating $Ca^{2+}$ permeability. Molecular dynamics simulations show that this glycine acts as a hinge, allowing for the C-terminal portion of the GluN1 M4 segment and the ion channel core to laterally expand. These results have strong implications for how neurological disorders are caused at the ion channel level and implicate the M4 as a major conduit for modulation of ion channel function.

## Results

**Disease-associated mutations are prominent at glycines**. At present, 23 patients with neurological disorders have been identified with de novo missense mutations in the M4 segments of NMDAR subunits (Fig. 1). The clinical phenotype for these patients is typically severe, encompassing epileptic encephalopathies and intellectual disabilities (Supplementary Table 1). For the majority of patients, the mutations occur at glycines (15 out of 23 patients). In transmembrane regions, the most common missense mutation associated with disease is glycine-to-arginine[18,19]. Glycines often participate in critical structural roles in transmembrane segments, including acting as a notch for transmembrane segment interactions[20] and to provide flexibility to transmembrane segments[21]. Substitution of a charged side chain for a glycine can strongly disrupt these functions.

Of the 15 patients with missense mutations at glycines, eight carry mutations to charged side chains: six are arginines (R) [GluN1(G815R), GluN1(G827R)] and two are glutamate (E) [GluN2B(G820E), GluN2B(G826E)]. However, for a glycine at the external end of M4, a number of patients have mutations that have less dramatic changes in the nature of the side chain, from glycine to alanine (A) or valine (V). This glycine, G815 in GluN1, G819 in GluN2A, and G820 in GluN2B, is one of only three positions completely conserved across all mammalian iGluR M4 segments (Supplementary Figure 1a) and also shows considerable conservation across many different species (Supplementary Figure 1b). That less dramatic missense mutations (e.g., alanine) can have a clinical phenotype and its conservation suggest that this glycine plays an important role in iGluR function.

**M4 missense mutations dramatically alter receptor gating**. The association of NMDAR missense mutations with neurological diseases could reflect alterations in the rules of NMDAR subunit assembly, trafficking[22], interaction with other intracellular or extracellular proteins, and/or receptor functional properties, such as gating, ion permeation, or channel block[23,24]. The majority of missense mutations in the M4 segment are positioned at the external ends, which participate strongly in gating and make no apparent contribution to a cell biological function (e.g., assembly)[16]. We therefore characterized the effect of many of these missense mutations on receptor gating (Fig. 2 and Supplementary Figure 2) using an external solution that approximates physiological conditions (see Methods).

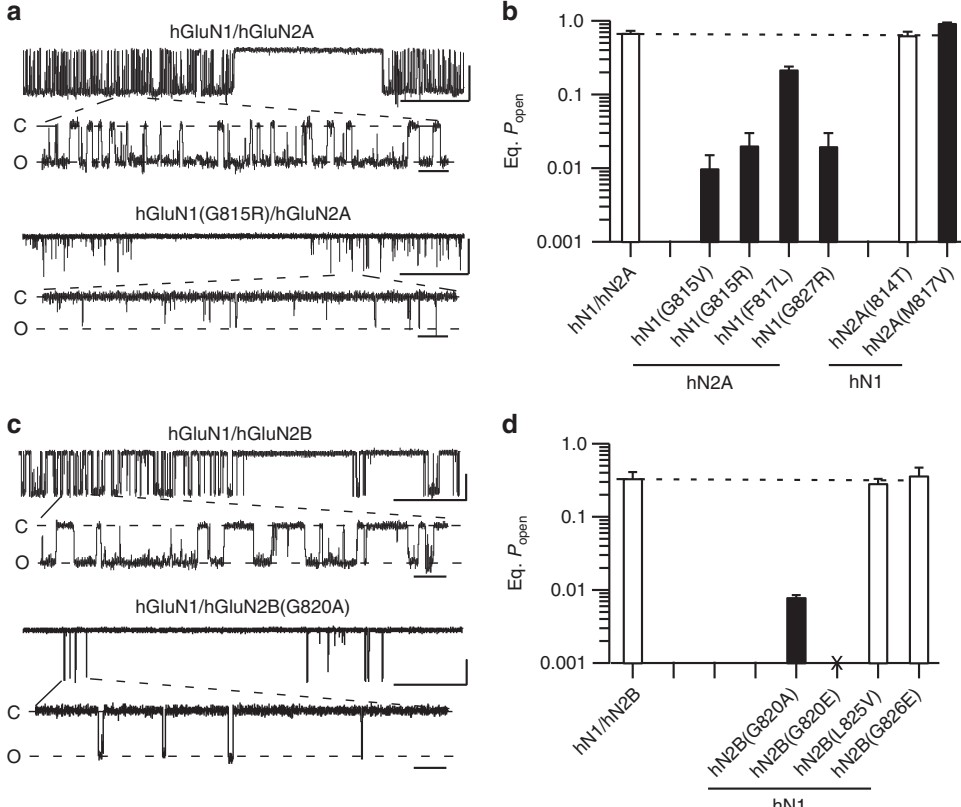

**Fig. 3** Single channel recordings of missense mutations in M4 segments. **a**, **c** Example single channel recordings of hGluN1/hGluN2A (**a**) or hGluN1/hGluN2B (**c**) or the same receptor containing missense mutations at the conserved glycine. Recordings were performed in the on-cell configuration (holding potential, +100 mV). Downward deflections are inward currents. For each construct, the top half shows a low-resolution (filtered at 1 kHz) and the bottom half a higher resolution portion of same record (3 kHz). Scale bars: 5 pAs for all; time base is 500 ms (upper trace for each construct) and 20 ms (lower trace for each construct). **b**, **d** Equilibrium open probability (eq. $P_{open}$) (mean ± SEM) ($n > 4$ patches) for hGluN1/hGluN2A (**b**) or hGluN1/hGluN2B (**d**) (Supplementary Table 3). Solid bars indicate values significantly different from wild type ($p < 0.05$, $t$-test). For GluN1/GluN2B(G820E), we could not detect glutamate-activated current either in on-cell patches or whole-cell mode

A critical determinant of synaptic NMDARs is their slow deactivation time course, which is GluN2-subunit specific[25–27]. Currents in human GluN1/GluN2A (hN1/hN2A) (Fig. 2a, left) and GluN1/GluN2B (hN1/hN2B) (Fig. 2c, top), induced by a brief glutamate application as occurs at synapses, decay slowly with the time courses best fit by double exponentials (weighted $\tau$s, 81 ± 2 ms, $n = 11$ and 660 ± 30 ms, $n = 8$, respectively) (mean ± SEM, $n$ = number of recordings) (Fig. 2b, d; Supplementary Table 2). As expected, the decay was much slower for GluN2B-containing receptors[26].

Several of the missense mutations had no significant effect on deactivation times (Fig. 2b, d, open bars), whereas two, hGluN2A (M817V)[17] and hGluN2B(G826E), significantly slowed deactivation. However, the most prominent effects of the M4 missense mutations was a significant speeding of deactivation, which was notable for mutations at the conserved glycine in hGluN1, G815V (35 ± 4 ms, $n = 5$) and G815R (39 ± 2 ms, $n = 6$) (Fig. 2b, filled bars) and in hGluN2B, G820A (205.2 ± 12 ms, $n = 7$) (Fig. 2d, filled bars). hGluN1(G815R) reduced the deactivation times by about half whether in the hGluN2A or hGluN2B background. We could not detect glutamate-activated whole-cell current for hGluN1(G827R)/hGluN2A or hGluN1/hGluN2B(G820E), either with brief or more sustained (Supplementary Figure 2) applications.

Although, the missense mutations could have additional effects on receptor function (e.g., permeation) or cell biology, these

strong effects on receptor deactivation (Fig. 2) as well as those on desensitization (Supplementary Figure 2), presumably contribute to their associated clinical phenotype. Here we will focus on how these mutations impact receptor function.

**Subsets of missense mutations strongly alter single channel activity.** To begin to understand how the missense mutations impact receptor gating, we recorded single channel activity in the on-cell mode (Fig. 3a–d; Supplementary Figure 3 and Supplementary Table 3). To obtain more mechanistic insight, we used an external solution that optimizes NMDAR activity (see Methods). Wild-type hGluN1/hGluN2A showed a mean equilibrium open probability (eq. $P_{open}$) of 0.69 ± 0.04, $n = 6$ (mean ± SEM, $n$ = number of single channel patches) (Fig. 3a, c) whereas hGluN1/hGluN2B showed a reduced eq. $P_{open}$ (0.34 ± 0.07, $n = 5$) (Fig. 3b, d, Supplementary Table 3), as found previously[28,29]. Certain missense mutations had no effect on activity (Fig. 3b, d, open bars), whereas one, GluN2A(M817V), showed a significant potentiation[17].

On the other hand, most tested missense mutations reduced single channel activity with the strongest effects occurring at glycines. At the conserved glycine, G815 in GluN1 and G820 in GluN2B, eq. $P_{open}$ was <0.02 (G815V, 0.01 ± 0.005, $n = 4$; G815R, 0.02 ± 0.01, $n = 5$; and G820A, 0.008 ± 0.0005, $n = 4$). For GluN1/GluN2B(G820E), which showed no whole-cell

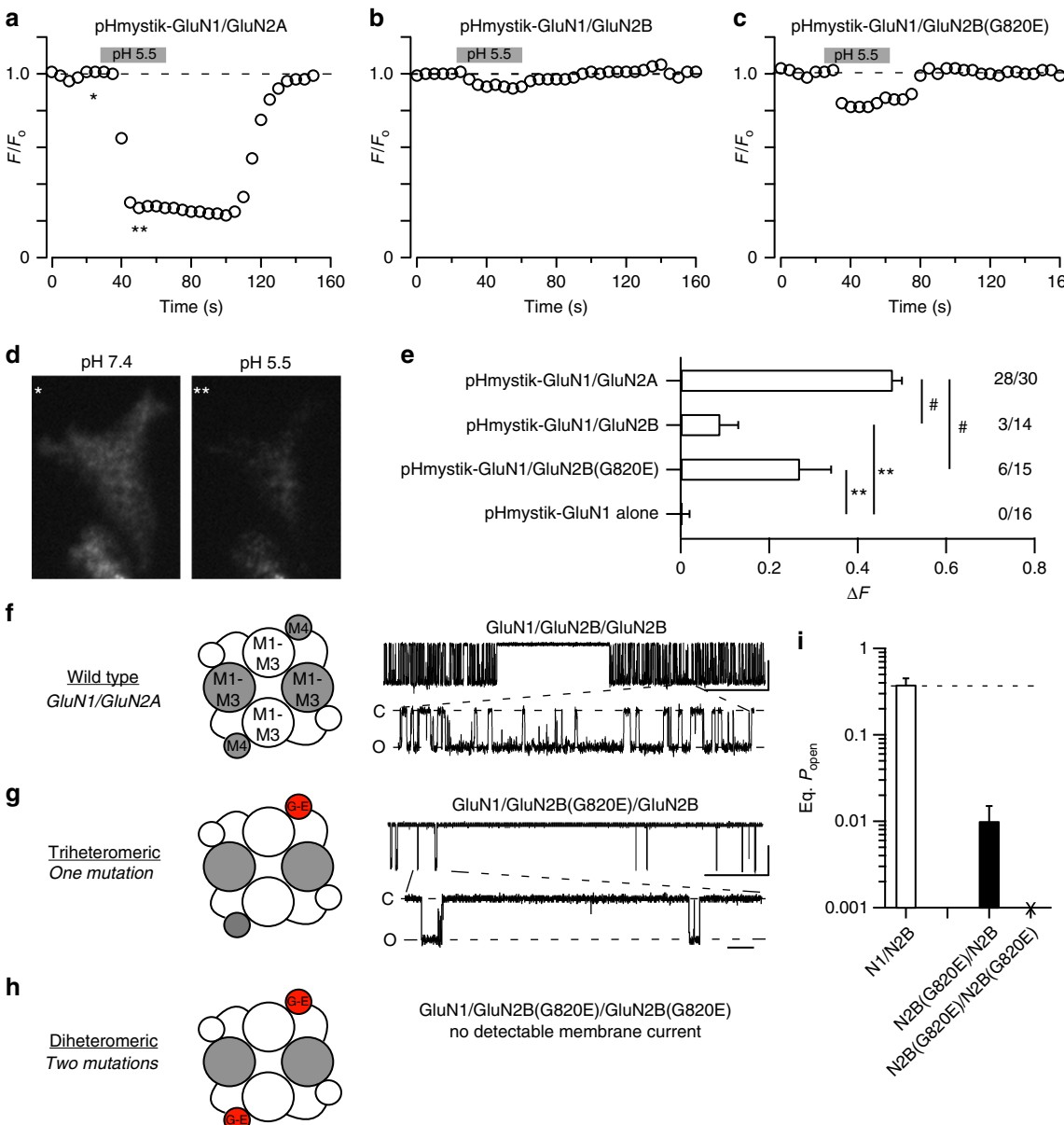

**Fig. 4** The diheteromeric missense mutation of a conserved glycine is pore dead. **a–c** Assaying surface expression using a pH-sensitive GFP[16] indicates that GluN1/GluN2B(G820E) is expressed on the membrane. GFP intensity as the extracellular solution pH was changed from 7.4 to 5.5 (gray bar, 30 s) and back again. **d** Representative cell images are from the time-points labeled in **a** with asterisks. Image pH 7.4 (single asterisk) is baseline fluorescence ($F_o$), whereas image pH 5.5 (double asterisks) is test fluorescence ($F_{test}$). The change in fluorescence ($\Delta F = F_o - F_{test}$) was used as an index of surface expression. Sampling rate, 5 s. The strong fluorescent background for GluN2B constructs presumably reflects subunits trapped in endoplasmic reticulum. **e** Changes in cell fluorescence at low pH ($\Delta F$). Significant differences from pHmystick-GluN1/GluN2A and pHmystik-GluN1 alone are indicated by # and **, respectively, ($p < 0.05$, t-test). The numbers (far right) indicate the number of cells that showed detectable changes in fluorescence relative to the total number of cells tested. **f–h** Single channel recordings of NMDAR constructs containing the coiled-coiled domain in C-terminus[30], either wild type (**f**), the triheteromeric containing a single copy of G820E (**g**) or the diheteromeric containing 2 copies of G820E (**h**), which does not show detectable glutamate-activated current (Figs. 2 and 3), though it expresses on the membrane. The triheteromeric receptor shows detectable single channel activity. Scale bars: 5 pAs for all; time base is 500 ms (upper trace for each construct) and 20 ms (lower trace for each construct). **i** Mean ( ± SEM) showing eq. $P_{open}$ for wild type ($n = 4$) and triheteromeric GluN1/GluN2B/GluN2B(G820E) ($n = 3$)

current amplitudes, no single channel activity could be detected (Fig. 3d; Supplementary Table 3). We detected single channel activity for a missense mutation at a glycine positioned more intracellularly, GluN1(G827R), for which we could not detect whole-cell current. Again, like other missense mutations at glycines, gating in GluN1(G827R) was strongly disrupted ($0.02 \pm 0.01$, $n = 4$) (Fig. 3b). The lack of detectable whole-cell

currents for GluN1(G827R) probably reflects that C-terminal positions in M4, especially this glycine, impact receptor surface expression[16].

**A missense mutation at the conserved glycine is pore dead.** The lack of glutamate-activated current for GluN2B(G820E), either in

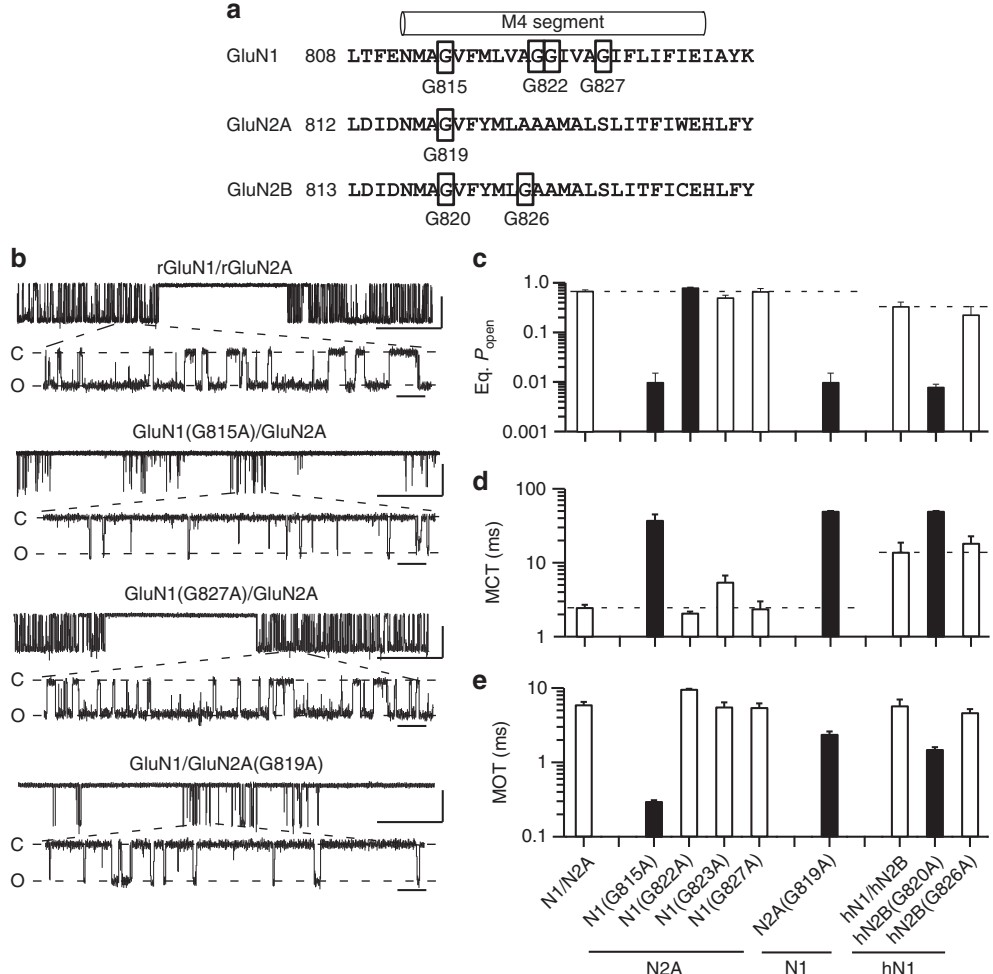

**Fig. 5** Alanine substitutions of glycine in the M4 segments reveal that the conserved glycine has a unique functional role in gating. **a** Sequence of M4 segments and adjacent regions highlighting glycines. **b** Example single channel recordings in the on-cell configuration of wild type rat GluN1/GluN2A and various receptors containing glycine-to-alanine substitutions. Recordings displayed as in Fig. 3a. Scale bars: 5 pAs for all; time base is 500 ms (upper trace for each construct) and 20 ms (lower trace for each construct). **c**–**e** Mean ± SEM ($n > 4$ patches) showing equilibrium open probability (eq. $P_{open}$) (**c**), mean closed time (MCT) (**d**), and mean open time (MOT) (**e**) for alanine substitutions of glycines in the M4 segments of rat GluN1 and GluN2A and human GluN1 (hGluN1) and GluN2B (hGluN2B) (Supplementary Table 4). Solid bars indicate values significantly different from their respective wild types ($p < 0.01$, t-test)

the whole-cell or on-cell configuration (Figs. 2 and 3), could reflect disrupted cell biology, either assembly and/or trafficking, or a complete lack of gating. To test these alternatives, we measured surface expression of GluN2B(G820E) using pHmystik, a pH-sensitive GFP, and TIRF microscopy[16]. In contrast to wild-type GluN1/GluN2A (Fig. 4a), wild-type GluN1/GluN2B showed poor surface expression (Fig. 4b, e) with the strong background fluorescence presumably due to receptors trapped in the endoplasmic reticulum. In contrast, GluN1/GluN2B(G820E) showed robust surface expression (Fig. 4c, e). Thus, this missense mutation in GluN2B, G820E, at least in a diheteromeric form, is pore dead: it can get to the membrane surface but cannot gate, an effect found for other positions at the extracellular end of GluN2 M4[16].

To further test the importance of the conserved glycine in GluN2B, we performed a rescue experiment in which we recorded single channel activity from triheteromeric receptors, where the receptor contains a single copy of G820E (Fig. 4f–i). To generate triheteromeric receptors, we used NMDAR constructs having coiled-coiled domains[30] that permit only specific subunit combinations to reach the plasma membrane. The triheteromeric

form, GluN1/GluN2B/GluN2B(G820E), is functional (Fig. 4g), but like other missense mutations at this conserved glycine, gating is severely restricted (eq. $P_{open}$, $0.007 \pm 0.001$, $n = 3$) (Fig. 4i).

In summary, many of the missense mutations in the M4 segment strongly restrict NMDAR gating (Figs. 2, 3 and 4; Supplementary Figure 3). These effects are notable at glycines, but are most severe at a conserved glycine, leading to in the diheteromeric form a receptor that transits to the membrane but is unable to gate (Fig. 4a–e).

**The conserved glycine in the M4s is key for pore opening.** The M4 segments must be displaced for efficient pore opening to occur[31]. One possible explanation for the effect of missense mutations at glycines in the various M4 segments is that these glycines provide local flexibility for the gating process to proceed. To test this possibility, we substituted each individual glycine in the M4 segments of GluN1, GluN2A, or GluN2B (Fig. 5a), including those not associated with missense mutations, with alanine, which would constrain local flexibility[32] while not greatly altering local interactions as might occur with charged side chains. We measured single-channel activity for these constructs (Fig. 5b–e).

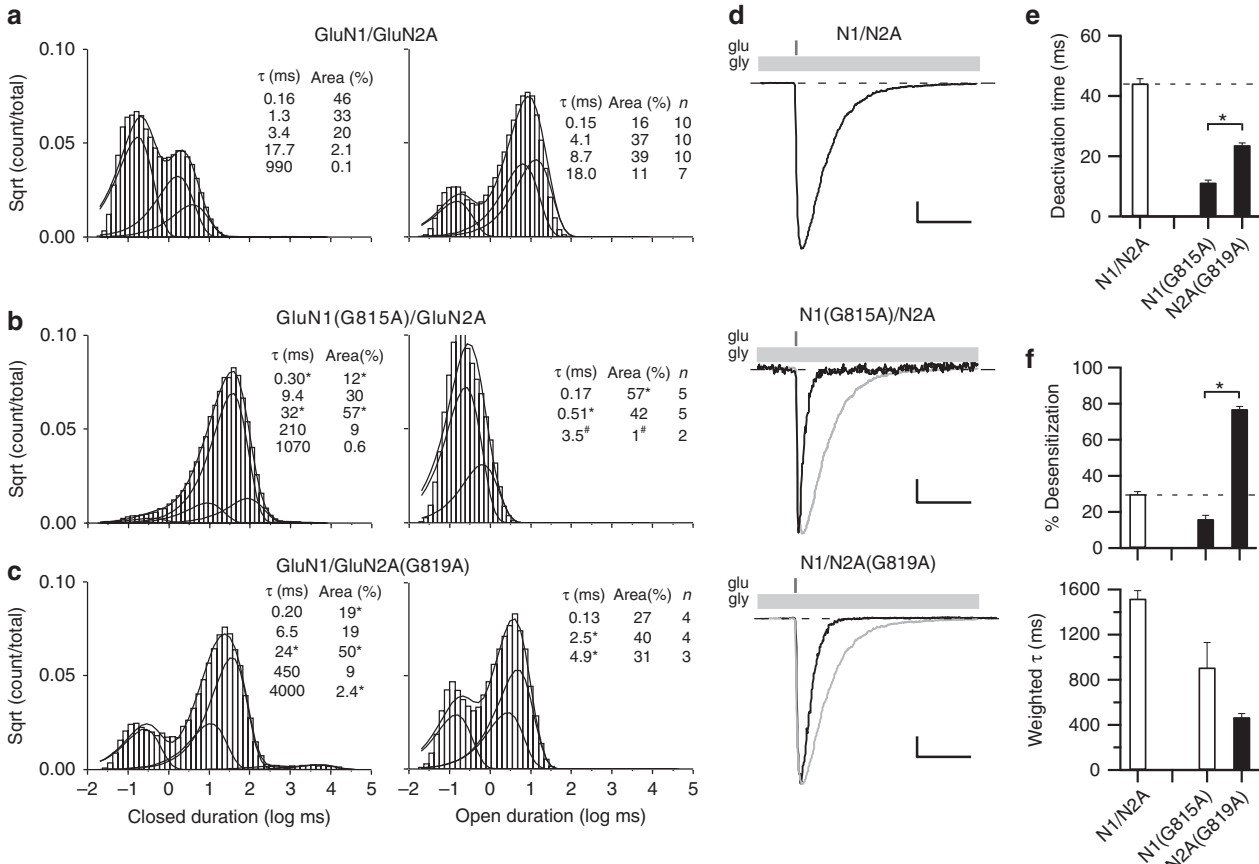

**Fig. 6** Constraining the conserved glycine prevents entry into long-lived open states and speeds deactivation. **a–c** Closed (left) and open (right) time histograms for wild type (**a**) or for receptors containing an alanine substitution at the conserved glycine in GluN1 (**b**) or GluN2A (**c**). All closed time histograms were best fit with 5 exponentials (Supplementary Table 5), whereas open time histograms were best fit typically with 4 (wild type), 2 (GluN1 (G815A)), or 3 (GluN2A(G819A)) exponentials (Supplementary Table 6) (see Methods). Smooth lines are associated exponential fits. Insets, mean closed and open state durations ($\tau$, ms), occupancies ($\alpha$, %), and for open time distributions how many recordings contained that open time (Supplementary Tables 5 & 6). For GluN1(G815A), the longest-lived open state (3.5 ms) was identified in only 2 out of 5 recordings; # indicates that we did not do statistics on this state. **d** Whole-cell currents in response to a 2 ms application of glutamate (1 mM, gray bars) applied in the continuous presence of glycine (light gray bars). Recordings were made and analyzed as in Fig. 2a but included extracellular 0.05 mM EDTA to remove effects of $Zn^{2+}$ as done for single channel recordings. Scale bars: 300 pA, wild type; 10 pA, N1(G815A); 100 pA, N2A(G819A); time base is 100 ms for all. **e**, **f** Mean ± SEM ($n > 5$) showing deactivation rates (**e**) or desensitization properties (**f**) for constructs shown in **d**. Solid bars indicate values significantly different from wild type, whereas asterisks indicate values different from each other ($p < 0.05$, ANOVA,Tukey)

Many of the glycine-to-alanine substitutions in GluN1, GluN2A, or GluN2B M4s either had no effect (open bars) or weakly altered eq. $P_{open}$ (G822 in GluN1). Notably, a charged missense mutation in a GluN1 glycine (G827R) had a dramatic effect on gating (Fig. 3b), but the subtler alanine had no significant effect (Fig. 5c; Supplementary Table 4). In contrast, alanine substitutions at the conserved glycine in all three subunits, GluN1(G815A), GluN2A(G819A), and hGluN2B (G820A), dramatically altered gating, reducing eq. $P_{open}$ to less than 0.01. Thus, the conserved glycine plays a unique role irreplaceable by any other side chain.

Alanine substitutions at the conserved glycines dramatically reduced receptor gating by significantly altering mean closed time (MCT) and mean open time (MOT) (Fig. 5d, e; Supplementary Table 4). In addition, the alanine substitution in GluN1 reduced single-channel activity to a similar extent as the charged (G815R) missense mutation (Fig. 3b) or when tryptophan was substituted[16].

**The conserved glycine permits slow deactivation.** To further address the impact of the conserved glycines to gating, we

characterized closed and open time distributions of the glycine-to-alanine substitutions in GluN1 and GluN2A (Fig. 6a–c). We focused functional experiments on GluN2A rather than GluN2B because of its more robust eq. $P_{open}$.

Our wild-type recordings were best described by 5 closed states (Fig. 6a, left; Supplementary Table 5)[33], as were the recordings for alanine substitutions at the conserved glycine position in GluN1 (Fig. 6b, left) and GluN2A (Fig. 6c, left). Their time course and specific duration, however, showed significant variation (Supplementary Table 5).

Wild-type GluN1/GluN2A shows modal gating consisting of a brief ($O_1$) and three longer lived ($O_2$, $O_3$, and $O_4$) open states[28]. For our recordings, open time distributions of wild type were also generally best described by 4 open states (Fig. 6a, right; Supplementary Table 6). In contrast, the open time distributions for glycine-to-alanine substitutions were at best only fit by 3 components, with a long-lived open state completely absent (Fig. 6b, c, right). Further, while GluN1(G815A) and GluN2A (G819A) visited a third open state, the durations were shorter (3.5 ms, $n = 2$ and 4.9 ± 1.0 ms, $n = 3$, respectively) compared to 8.7 ± 0.6 ms, $n = 10$ for wild type). Finally, GluN1(G815A) visited

a third open state extremely rarely (2 out of 5 recordings), around 1% of the time compared to $39 \pm 10\%$ for wild type, highlighting the more prominent role of the GluN1 conserved glycine.

The kinetics of the NMDAR single-channel activity mediate the slow macroscopic deactivation time courses characteristic of synaptic NMDARs[25,34]. We therefore directly compared deactivation rates for GluN1/GluN2A, GluN1(G815A)/GluN2A, and GluN1/GluN2A(G819A) under conditions comparable to those done for single channel recordings (extracellular 0.05 mM EDTA) (Fig. 6d). Consistent with the reduced long open times, the deactivation times for GluN1(G815A) ($11.3 \pm 0.8$ ms, $n = 5$) and GluN2A(G819A) ($23.7 \pm 0.7$ ms, $n = 5$) were significantly faster than that for wild type ($44.2 \pm 1.5$ ms, $n = 5$). Paralleling the results for MOTs, deactivation for GluN1(G815A) was significantly faster than that for GluN2A(G819A) (Fig. 6e).

One potential explanation for the reduced MOTs and speeding of deactivation rates is that alanine substitutions of the conserved glycines disrupt the stability of the clam-shell closed, agonist-bound LBD conformation. To test this possibility, we used cysteine cross-links in the LBD that lock the clam-shell closed (Supplementary Figure 4). The eq. $P_{\text{open}}$ (Supplementary Figure 4b) and MOT (Supplementary Figure 4d) for glycine-to-alanine substitutions in either GluN1 or GluN2A was indistinguishable with or without the clam-shell locked close, arguing against any strong effect of the conserved glycine on LBD conformation. We also tested glycine deactivation (Supplementary Figure 5) and it too was significantly faster for both GluN1 (G815A) and GluN2A(G819A) relative to wild type (Supplementary Figure 5c). Finally, desensitization properties for the glycine-to-alanine receptors do not appear to contribute strongly to differences in deactivation rates (Fig. 6f).

In summary, alanine substitution of the conserved glycine in either GluN1 or GluN2A attenuates long-lived open states and speeds receptor deactivation, with these effects significantly more prominent in GluN1. These effects are primarily due to this conserved glycine regulating the ion channel conformation as opposed to the LBD conformation.

**The conserved glycine maintains the open postures of the M4s.** At present there are no open state structures available for NMDARs. To understand how alanine substitutions of the conserved glycines might impact receptor function, we used homology modeling and molecular dynamic (MD) simulations. Based on our previous study[35], we built a homology model for the GluN1/GluN2B open state using the AMPAR open structure 5WEO[36] as template, introduced glycine-to-alanines, and ran simulations for 500 ns each for wild type, GluN1(G815A)/GluN2B, and GluN1/GluN2B(G820A) (Supplementary Figure 6 and Fig. 7). Partly due to uncertainty of the wild-type open model, we focused our analysis on the relative difference between glycine-to-alanine mutants and wild type. We refer to this approach as comparative MD simulations.

During the MD simulations, the Cα RMSDs of the entire TMD from the homology model stabilized after about 200 ns for all constructs (Supplementary Figure 6a). We therefore used the 250–500 ns portion of the MD trajectories to calculate average properties. For wild type the TMD Cα RMSD stabilized at $2.5 \pm 0.2$ Å (mean value $\pm$ SD) (Supplementary Figure 6b). For both alanine substitutions, the TMD Cα RMSDs increased and, paralleling the impact on MOT, the value was higher for GuN1 (G815A) ($3.1 \pm 0.1$ Å) compared to GluN2B(G820A) ($2.7 \pm 0.2$ Å). The Cα RMSDs for the TMD core (M1–M3) and the M4 segments showed similar trends (Fig. 7a) though with greater changes over time for the M4 segments (Fig. 7b). Thus, relative to wild type, introducing alanine at the conserved glycine in either

GluN1 or GluN2B alters not only M4 but the entire TMD structure, with these effects stronger for GluN1(G815A).

For wild type, a notable feature was the splaying of the C-termini of the M4 segments, with the hinge or pivot point occurring around the conserved glycine (Fig. 7c, d). In contrast, this splaying was considerably more limited for the GluN1 M4s in GluN1(G815A) (Fig. 7e, f), while a less drastic change was observed for the GluN2 M4s in GluN2B(G820A) (Fig. 7h, i). To quantify this effect, we measured the M4 Cα displacements of GluN1(G815A) (Fig. 7g) and GluN2B(G820A) (Fig. 7j) from wild type. The M4 N-terminal portions of the mutants and wild type remained closely aligned. However, the displacements gradually increased toward the C-termini, where the values were much greater for GluN1 subunits in GluN1(G815A) (Fig. 7g; A subunit, 5.9 Å; C, 8.9 Å) than for GluN2 subunits in GluN2B(G820A) (Fig. 7j; B subunit, 0.9 Å; D, 5.3 Å). Again, these changes in the positioning of the M4 segments parallel the effect of the alanine substitutions on the stability of the open channel (Fig. 6a–c).

**The conserved glycine permits high $Ca^{2+}$ permeability.** Given the differences in C-terminal splaying between the constructs (Fig. 7), we characterized how the glycine-to-alanine mutations might impact the C-terminal portion of the permeation pathway as defined by the M2 pore helix (Fig. 8a, b). For wild type, the average pore radius over the M2 helix is 2.0 Å (minimum at 1.6 Å) (Fig. 8a, b, gold). In comparison, for GluN1(G815A), the pore is narrower, averaging around 1.6 Å (minimum, 1.2 Å) (Fig. 8a, right), whereas for GluN2B(G820A), it is unchanged 2.1 Å (minimum at 1.8 Å) (Fig. 8b, right).

The M2 helix regulates $Ca^{2+}$ permeability in NMDARs[8]. We therefore measured relative $Ca^{2+}$ permeability using changes in reversal potential (Fig. 8c, d)[37]. Wild-type GluN1/GluN2A, ongoing from 0 to 10 mM $Ca^{2+}$, showed a change of reversal potential ($\Delta E_{\text{rev}}$) of about 10 mV ($9.5 \pm 0.9$ mV, $n = 7$), yielding a $P_{\text{Ca}}/P_{\text{Na}}$ of 4 ($4.1 \pm 0.6$). For GluN1(G815A), where the M2 pore size was reduced in the MD trajectory, the reversal potential change was significantly reduced ($5.7 \pm 0.5$ mV, $n = 8$), corresponding to a decrease in $P_{\text{Ca}}/P_{\text{Na}}$ by about half ($2.0 \pm 0.2$, $n = 8$). In contrast, GluN1/GluN2A(G819A), where pore size was not altered, $P_{\text{Ca}}/P_{\text{Na}}$ was not significantly changed ($3.7 \pm 0.6$, $n = 7$).

We also measured $Ca^{2+}$ permeability for a missense mutation, G815R, at the conserved glycine in GluN1. For this construct, $Ca^{2+}$ permeability was altered to the same extent as the alanine (Fig. 8d) indicating that electrostatics do not contribute to the role of the GluN1 conserved glycine in modifying $Ca^{2+}$ permeability.

## Discussion

A key function of NMDARs at synapses is to mediate a time-dependent $Ca^{2+}$ influx that defines synaptic strength and dynamics of integration[8,27]. These features are dependent on the slow deactivation time course of NMDARs in response to transient glutamate released presynaptically and a high $Ca^{2+}$ permeability. Here, using a combination of functional experiments and MD simulations, our data on clinically relevant mutations show that the C-terminal splaying of the M4 segments, mainly in the GluN1 subunit, regulates both deactivation and high $Ca^{2+}$ permeability (Fig. 9). These observations have multiple implications for iGluR gating and permeation, including the role of NMDARs in disease and modulation via the C-terminal domain.

The NMDAR open state model was built using homology modeling of an AMPAR open structure[36] (see Methods). Although this model (Fig. 7c, d) may capture the open state of wild-type NMDARs, we did not test this rigorously in the present

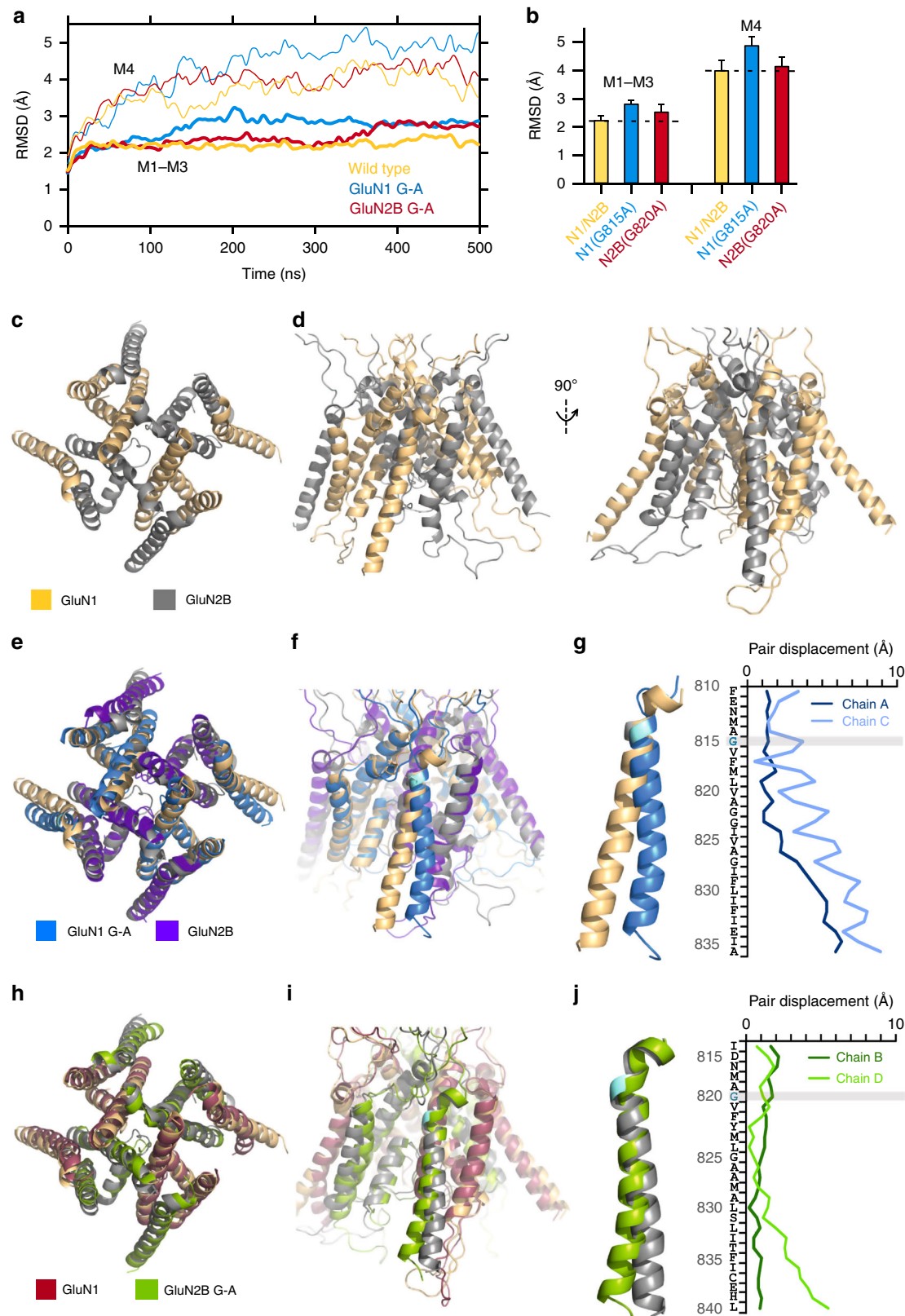

context. Rather, we focused on the relative difference between wild type and constructs containing glycine-to-alanine substitutions either in GluN1 (Fig. 7e–g) or GluN2B (Fig. 7h–j) in an approach that we term comparative MD simulations. Both of these constructs, relative to wild type, showed deviations for all

transmembrane segments with these effects consistently stronger for GluN1(G815A) (Fig. 7b and Supplementary Figure 6b). The most notable difference between wild type and the glycine-to-alanine constructs was in the M4 segments, as clearly revealed by GluN1(G815A), where the conserved glycine maintains the

**Fig. 7** MD simulations show that constraining the conserved glycine in GluN1 prevents C-terminal expansion of the M4 as observed in wild type. **a** Cα RMSDs of the ion channel cores (M1–M3, thick lines) and M4s (thin lines) as a function of simulation time for open state models: wild-type GluN1/GluN2B, light orange; GluN1(G815A)/GluN2B, light blue; GluN1/GluN2B(G820A), maroon (see Methods). **b** Average RMSD values (250–500 ns). Error bars are standard deviations. **c, d** Open state of wild-type GluN1/GluN2B TMD shown either top down (**c**) or side views (**d**) of the GluN1 (left) or GluN2B (right) M4s. All displayed structures are snapshots close to the average structures calculated over 250–500 ns. **e, f** GluN1(G815A) overlaid on wild type either from top down (**e**) or with a side view (**f**). **g** Displacements between GluN1(G815A) and wild type for M4 residues in the A and C subunits. **h, i** GluN2B (G820A) overlaid on wild type either from top down (**h**) or with a side view (**i**). **j** Displacements between GluN2B(G820A) and wild type for M4 residues in the B and D subunits

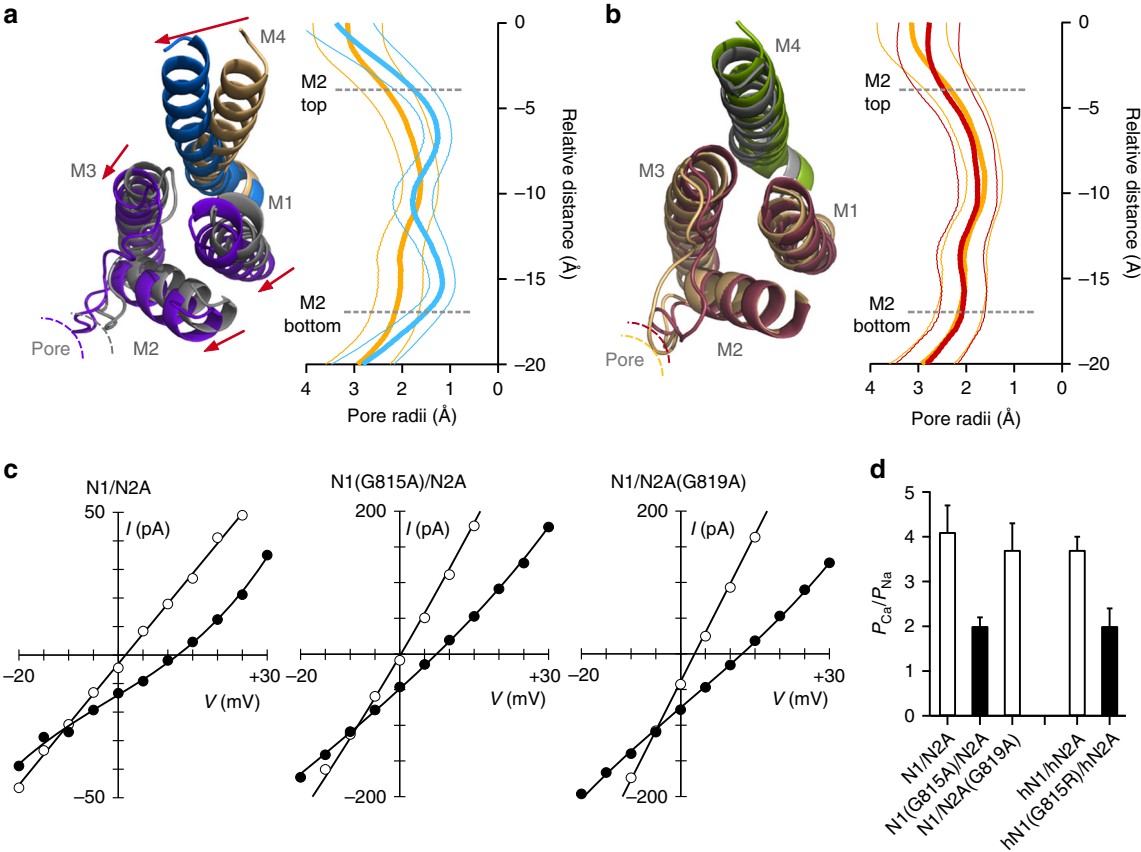

**Fig. 8** Constraining the conserved G in GluN1 reduces the dimensions of the selectivity filter and Ca$^{2+}$ permeability. **a** Left, Cytoplasmic view of model open state structures of wild type and GluN1(G815A). The constrained M4 collapses the adjacent GluN2B ion channel core (M1–M3) including the pore size as defined by the M2 loop. Right, Average pore radius along the channel axis for the M2 pore loop for wild type (gold) and GluN1(G815A) (light blue). '0' references the serine (S) in SYTANLAAF in M3. Thick lines are mean values and thinner lines the error bars, which were calculated using block averages based on a time block of 20 ns. **b** Same as **a** except construct is GluN2B(G820A) (maroon), which resulted in a less displacement of the M2 loop. **c** Current-voltage (IV) relationships in an external solution containing high Na$^+$ (140 mM) and 0 added Ca$^{2+}$ (open circles) or 10 mM Ca$^{2+}$ (solid circles). The 0 Ca$^{2+}$ IV is the average of that recorded before and after the 10 mM Ca$^{2+}$ recording. We used changes in reversal potentials ($\Delta E_{rev}$) to calculate $P_{Ca}/P_{Na}$[37]. **d** Mean ± SEM showing relative calcium permeability ($P_{Ca}/P_{Na}$) calculated from $\Delta E_{rev}$s for rat GluN1/GluN2A ($n = 7$), GluN1(G815A) ($n = 8$), or GluN2A(G819A) ($n = 7$) as well as human GluN1/GluN2A ($n = 11$) and the missense mutation GluN1(G815R) ($n = 7$). Values either are not (open bars) or are (solids bars) significantly different from each other ($p < 0.05$, ANOVA, Tukey)

bottom two-thirds of the helix in an open posture, leaving the upper one-third in close proximity to other TMD elements (Fig. 7g). We assume that the major effect of the glycine-to-alanine in GluN1 is to limit this splaying of M4, reducing the stability of long-lived open state (Fig. 9). The glycine-to-alanine substitution in GluN2B also alters the positioning of M4 relative to wild type (Fig. 7j), but this effect is small and may not reflect the major pathway by which the conserved glycine in GluN2 impacts receptor function.

One prediction arising from the comparative MD simulations is that the glycine-to-alanine in GluN1 but not GluN2 restricts

the pore diameter of the M2 pore loop (Fig. 8a, b). Supporting this observation is that Ca$^{2+}$ permeation in GluN1(G815A) but not GluN2A(G819A) was significantly reduced (Fig. 8c, d). This result supports the relative differences in the MD simulations and also indicates that pore expansion impacts the high Ca$^{2+}$ influx mediated by NMDARs.

That glycine-to-alanine substitutions at neighboring positions in the M4 segments had no significant effects on receptor gating (Fig. 5) suggest that, rather than merely providing local flexibility to the M4 segments, the conserved glycines play unique roles irreplaceable by any other side chain. In part it may reflect that

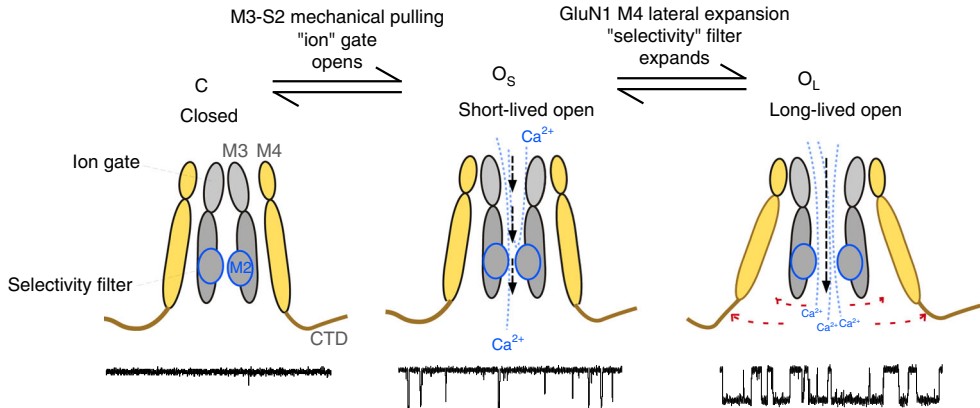

**Fig. 9** Expansion of the inner pore permitted mainly by the GluN1 M4 conserved glycine facilitates NMDAR gating and $Ca^{2+}$ permeation. Cartoon of NMDAR pore structure, including the influence of the GluN1 M4 on open state stability and selectivity filter integrity. Subunits are colored light orange (GluN1) and gray (GluN2)

the relative positioning of the glycine in the M4 helix is critical, with structural elements positioned external to the glycine being involved in critical interactions[23] and those internal to the glycine requiring flexibility[16]. In addition, the open state model suggests a Cα hydrogen bond that connects the S1-M1 directly to the conserved glycine in the M4[35], which could also alter receptor gating.

We also envision that the M4 is passively, not actively, splaying in response to pore opening. Consistent with this idea, specific side chain interactions of the C-terminal two-thirds of the M4 helix appear unimportant to gating[16] as would be expected if it acted permissively.

Given the high number of patients identified with mutations in the conserved glycine relative to other M4 transmembrane segment mutations, it is unlikely that mutations at this site are merely epiphenomena. Indeed, such drastic alterations in NMDAR structure and subsequent dysfunction imply a causal relationship. One therefore asks what aspect of this dysfunction plays the dominant role in the pathogenesis of these disorders. Notably, most M4 missense mutations that we measured had reduced channel activity characterized by a destabilized open state (Fig. 3). Though a reduction in excitatory glutamatergic activity leading to epilepsy may seem counterintuitive, it most likely changes the balance of excitation and inhibition in the nervous system.

The most identifiable pathology amongst these mutations is epilepsy. However, a dysfunction common to nearly all M4 missense mutations is intellectual disability (Supplementary Table 1), suggesting that the problem may lie in development. The GluN2B subunit, which carries numerous mutations at the conserved glycine, is most highly expressed before birth[27]. GluN1, being an obligate subunit, is expressed at all stages of development. Interestingly, the absence of any conserved G mutants in GluN2A, a subunit that is primarily expressed in the adult brain, suggest that pathogenesis of these disorders strongly involves the NMDARs role in the developing brain.

The M4 transmembrane helix in iGluRs is the most peripheral transmembrane segment in the iGluR ion channel, interacts with lipids, and is attached to the regulatory C-terminal domain. Lipids[38] and post-translational modifications of the C-terminal domain[39] can modulate ion channel gating and $Ca^{2+}$ permeation, but how they change receptor function is unknown. We hypothesize that a major pathway by which they act is by regulating the permissiveness of the M4. In addition, agonist binding in NMDARs can induce signaling independent of ion channel opening, including priming them for

internalization[40] and metabotropic actions[41]. Given that the M4s connect the external LBD and the internal CTD and that their displacements would not directly open the channel, the M4 segments could represent the pathway for such transmembrane signaling. Future experiments will be needed to directly test these ideas.

## Methods

**Molecular biology and cell culture**. All manipulations were made in rat GluN1 (GluN1a) (NCBI Protein database accession no. P35439), GluN2A (Q00959), or GluA2 (P19491) subunits or human GluN1 (hGluN1; Q05586), GluN2A (hGluN2A; Q12879), or GluN2B (hGluN2B; Q13224) subunits. Unless otherwise noted constructs tested were rat. In all instances, numbering included the signal peptide (GluN1, 18 residues; GluN2A, 19 residues; GluN2B, 26 residues; GluA2, 21 residues). Mutations were generated via QuickChange site-directed mutagenesis (Agilent) with XL1-Blue super-competent cells.

Cell culture and transfection: For details on cell culture and transfection see Amin et al., 2017. Briefly, human embryonic kidney 293 (HEK293) or HEK293T cells were grown in Dulbecco's modified Eagle's medium (DMEM), supplemented with 10% fetal bovine serum (FBS), for 24 h prior to transfection. Non-tagged cDNA constructs were co-transfected into HEK293 cells along with a separate pEGFP-Cl vector at a ratio of 4.5:4.5:1 (N1/N2/EGFP for NMDARs) using X-tremeGene HP (Roche). GFP-tagged constructs were transfected at a 1:1 ratio. To improve cell survivability, HEK293 cells were bathed in a media containing the NMDAR competitive antagonist APV (100 μM) and $Mg^{2+}$ (100 μM) (single channel experiments) or the transfection mixture was replaced 4 h after transfection with fresh 5% FBS-DMEM culture media containing APV (100 μM) and $Mg^{2+}$ (1 mM) (whole-cell and imaging experiments). All experiments were performed 18–48 h post transfection.

Triheteromeric expression system: NMDARs are obligate heterotetramers typically composed of 2 GluN1 and 2 GluN2 subunits. Co-expression of wild-type GluN1 and GluN2A and a subunit with a mutation [e.g., GluN2A(mutation)] would result in three populations of cell-surface expressed receptors: GluN1/ GluN2A (diheteromeric); GluN1/GluN2A/GluN2A(mutation) (triheteromeric); and GluN1/GluN2A(mutation) (diheteromeric). To restrict surface expression to defined triheteromeric receptors, we used the 'triheteromeric' system developed by the Traynelis lab[30], which is based on leucine zipper motifs from GABA$_B$ receptors and ER retention/retrieval motifs introduced into the C-terminal domain of NMDAR subunits. All constructs were kindly provided by Dr. Kasper Hansen (University of Montana).

**Molecular modeling and simulations**. Previously, we constructed an NMDAR open-state model by repacking the transmembrane helices[35]. The model was very similar to the GluA2 open-state structure (Protein Data Bank (PDB) entry 5WEO) [36], indicating that 5WEO could be a useful template for modeling the open state of the NMDAR TMD.

Homology modeling: We used homology modeling to build an initial model for the GluN1/GluN2B receptor in the open state (without the disordered C-terminal domain)[42]. We then introduced glycine-to-alanine mutations at the conserved position, producing the N1(G815A)/N2B and N1/N2B(G820A) mutants. For the wild-type open model, we chose PDB entry 5FXG[43] as the template for the extracellular amino-terminal domain (ATD) and LBD[35], and 5WEO[36] as the template for the TMD. The templates were both aligned to the

closed NMDAR structure 4TLM[44]. We then used Modeller 9.18[45] for homology modeling, including generation of missing residues.

Our previous study[35] suggested that interactions between the pre-M1 helix and the extracellular end of the M4 helix are important for stabilizing the open state. Among these interactions are special hydrogen bonds formed with the Cα atoms of the conserved glycines as donors. Here we introduced such hydrogen bonds into our initial open model, in a 14-ps fine-tuning simulation in vacuum using NAMD 2.11. In this simulation, all Cα atoms were restrained with a force constant of 5 kcal mol$^{-1}$ Å$^{-2}$ except for residues K544-Q559 in GluN1 and R540-S555 in GluN2B containing the pre-M1 helices. In addition, Cα atoms of G815 in GluN1 and G820 in GluN2B were restrained to a hydrogen-bonding distance of 3.2 Å from backbone carbonyl oxygen atoms of L551 in GluN1 and P547 on GluN2B, respectively, with a force constant of 0.1 kcal mol$^{-1}$ Å$^{-2}$. The G815A mutation in GluN1 and G820A mutation in GluN2B were made using VMD[46] to produce open models for the two mutants.

Comparative molecular dynamics (MD) simulations: Following our previous study[35], MD simulations were carried out for the TMD construct, consisting of residues L541-P670 and R794-A841 in GluN1 and M539-K669 and G799-F845 in GluN2B, truncated from our full-length models. The four LBD-proximal residues of the LBD-TMD linkers (L541-K544, I667-P670, R794-E797 in GluN1 and M537-R540, L666-K669, G799-H802 in GluN2B) were restraint to mimic the agonist-bound form of the LBD that maintains channel opening. The TMD construct was embedded in a membrane bilayer composed of 222 POPC lipids, and solvated by 25,000 water molecules and 0.15 M NaCl, using CHARMM-GUI[47]. CHARMM-GUI's standard 6-step equilibration protocol was followed to equilibrate the system. An additional equilibration step of 12 ns was used to ramp down the restraints on the Cα atoms except for the LBD-proximal linker residues. The final production run was 500 ns with restraints on the LBD-proximal linker Cα remain at 5 kcal mol$^{-1}$ Å$^{-2}$.

Starting from the CHARMM-GUI's equilibration step, simulations were performed using AMBER 14 on GPUs[48] with CHARMM36 force field[49]. The time step was 2 fs. Non-bonded interactions were cutoff at 8.0 Å, and the particle mesh Ewald method was used for long-range electrostatic interactions[48]. The NPT ensemble was used in simulation, with temperature controlled at 300 K by Langevin dynamics and pressure controlled at 1 bar by the Berendsen method[50].

Analysis: MD trajectories were saved once every 100 ps (5000 frames for each construct). Root-mean-square-deviations (RMSDs) were calculated using Cα atoms of either all the helical segments (M1–M4) or only M1–M3 in the ion channel core. In the latter case, the RMSD for M4 was also calculated using the structural alignment on M1–M3. The reference for all the RMSD calculations was the homology model. The aligned structures were used to calculate an average structure, which was then used to select a representative snapshot from each MD trajectory. The pore radii along the pore axis were calculated using the HOLE program[51] on every 10th of the saved frames.

**Macroscopic current recordings**. Macroscopic currents in the whole-cell mode or outside-out patches, isolated from HEK293 cells, were recorded at room temperature (20–23 °C) using an EPC-9 or EPC-10 amplifier with Patchmaster software (HEKA Elektronik, Lambrecht, Germany), digitized at 10 kHz and low-pass filtered at 2.8 kHz (−3 dB) using an 8 pole low pass Bessel filter. Patch microelectrodes were filled with our standard intracellular solution (in mM): 140 KCl, 10 HEPES, 1 BAPTA, pH 7.2 (KOH). Our standard extracellular solution consisted of (in mM): 140 NaCl, 1 CaCl$_2$, 10 HEPES, pH 7.2 (NaOH). Pipettes had resistances of 2–6 MΩ when filled with the pipette solution and measured in the standard Na$^+$ external solution. We did not use series resistance compensation nor did we correct for junction potentials. Currents were measured within 15 min of going whole cell.

External solutions were applied using a piezo-driven double barrel application system. The open tip response (10–90% rise time) of the application system was between 400 and 600 μs. For display, NMDAR currents were digitally refiltered at 500 Hz and resampled at 1 kHz.

Rates of deactivation and desensitization: To determine the rates of activation and deactivation, we applied glutamate for 2 ms at −70 mV in the whole-cell mode[52]. For activation times, we used the 10–90% rise time. Deactivation times (weighted τs) were derived by fitting the decay phase of currents with a double-exponential function. To determine the extent and rate of desensitization, we applied glutamate at −70 mV for 2.5 s in the whole-cell mode. Percent desensitization (% des) was calculated from the ratio of peak ($I_{peak}$) and steady-state ($I_{ss}$) current amplitudes: %des = $100 \times (1 - I_{ss}/I_{peak})$. Time constants of desensitization were determined by fitting the decaying phase of currents a double-exponential function. In some instances when current amplitudes were small, we averaged 3–12 records.

$Ca^{2+}$ permeability: To quantify $Ca^{2+}$ permeability, we measured changes in reversal potential, $\Delta E_{rev}$, for glutamate-activated currents on replacing a reference solution (140 mM NaCl, 10 mM HEPES, 0 added $Ca^{2+}$, pH 7.2, NaOH) with a test solution (the same solution but with added 10 mM CaCl$_2$). $Ca^{2+}$ permeability ratios ($P_{Ca}/P_{Na}$) were calculated using the Lewis equation[37]. We did not correct for activity coefficients.

**Single-channel recordings and analysis**. All single-channel recordings were performed in the on-cell configuration at steady state. The pipette solution, which mimicked extracellular agonist conditions, contained (in mM): 150 NaCl, 10 HEPES, 0.05 EDTA, 1 glutamate, and 0.1 glycine, pH 8.0 (NaOH). The high pH and EDTA were used to minimize proton and divalent mediated inhibitory effects, respectively[28]. Recording pipettes were pulled from thick-wall borosilicate capillary glass (Sutter Instruments) and fire-polished to final pipette resistances ranging from 5 to 30 MΩ when measured in the bath solution (with an applied positive pipette pressure of ~200 mbar). Cells were identified by GFP fluorescence and patched to resistances exceeding 1.5 GΩ. To elicit inward current amplitudes, we held the electrode voltage at +100 mV. Currents were recorded using a patch clamp amplifier (Axopatch 200B; Molecular Devices), filtered at 10 kHz (four-pole Bessel filter), and digitized at 40 kHz (ITC-16 interfaced with Patch-Master). Experiments ran for ~3–20 min to ensure a significant number of events for analysis.

NMDAR single channel analysis: Analysis of single-channel records was comparable to Talukder and Wollmuth[53]. Data were exported from PatchMaster to QuB. Processed data were idealized using the segmental k-means (SKM) algorithm in QuB with a dead time of 20 μs. Closed and open state fits were performed using the maximum interval likelihood (MIL) algorithm in QuB. For most recordings, we measured only single-channel amplitudes, mean closed time (MCT), and mean open time (MOT) as well as equilibrium open probability (eq. $P_o$).

For certain constructs, we explored in more detail the occupancy of different closed and open states. Kinetic models of NMDA receptor gating activation have been proposed to contain ~5 closed states and 3–4 open states[28,29]. For each individual record, state models with increasing closed (3–6) and open (2–4) states were constructed and fitted to the recordings until log-likelihood (LL) values improved by less than 10 LL units/added state or if the next added state showed 0% occupancy[53]. To verify single channels in patches, especially for those constructs with a low equilibrium $P_o$, we used statistical approaches[31].

**Assaying surface expression using pH-sensitive GFP**. HEK293T cells were imaged 24–48 h post transfection using a Nikon Ti Eclipse microscope equipped with the Nikon TIRF slider. Excitation used the 488 nm line from a fiber-coupled argon ion laser (Lasos, Jena, Germany)[16]. The fluorescence emission was collected with a Nikon ×60 1.45 NA oil-immersion TIRF objective and relayed to an iXon DU897 emCCD camera (Andor Technologies, South Windsor, CT). The fluorescence emission was transmitted using a dual band 488/561 TIRF filter cube with a GFP band pass of 525/50 (Chroma, Bellows Falls, VT). Cells were observed at 5 s intervals with 30–50 exposures collected for each field of view, which contained 1–4 fluorescent cells. Cells were imaged in either a bath solution at pH 7.4, consisting of (in mM): 140 NaCl, 10 HEPES, pH 7.4 (NaOH); or a bath solution at pH 5.5, consisting of (in mM): 140 NaCl, 30 2-(N-morpholino)ethanesulfonic acid (MES), pH 5.5 (HCl). These bathing solutions were exchanged using a continuous flow perfusion system.

Fluorescence intensity was quantified in ImageJ[54]. Cells were selected for analysis that were at least 50% isolated from other cells. The fluorescence intensity ($F$) of a cell of interest was calculated for each frame: $F = F_{cell} - F_{backgnd}$, where $F_{cell}$ is the mean fluorescence of the cell, defined by a polygon circumscribing it, and $F_{backgnd}$ is the mean fluorescence of an acellular region directly adjacent to the cell of interest. For display and analysis, we normalized fluorescence intensity to a baseline fluorescence ($F_o$), which was the mean $F$ just before the solution was changed from pH 7.4 to 5.5. The change in fluorescence ($\Delta F$) was defined as $\Delta F = F_o - F_{test}$, where $F_{test}$ is the fluorescence intensity taken 15–30 s after the solution was switched from pH 7.4 to 5.5. We defined 'detectable' surface expression as occurring when the fluorescence intensity decreased rapidly upon switching to pH 5.5 and returned to baseline with a return to pH 7.4. For some constructs, we detected both positive and negative (no detectable $\Delta F$); in these instances, we averaged surface expression only for those cells that were positive.

**Statistics**. Data analysis was performed using IgorPro, QuB, Excel, and MiniTab 18. All average values are presented as mean ± SEM. The number of replicates is indicated in the figure legend or in a table associated with the figure. Often, we were only interested in whether outcomes were statistically different from that for wild type. In these instances, we used an unpaired two-tailed Student's t-test to test for significant differences. An analysis of variance (ANOVA), followed by the Tukey or Dunnet's test, was used for multiple comparisons. Unless otherwise noted, statistical significance was set at $p < 0.05$.

## Data availability
Igor and/or Excel files of all raw data are available upon request.

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

## Acknowledgements

We thank Dr. Rashek Kazi and Kelvin Chan for helpful discussions and/or comments on the manuscript, Drs. Stephen Traynelis and Hongjie Yuan (Emory University) for generously sharing human NMDAR subunits, Dr. Kasper Hansen (University of Montana) for supplying triheteromeric constructs, and Gabrielle Moody for assisting with Supplementary Table 1. Dr. Steve Smith (Stony Brook) provided invaluable insights about the importance of glycines in transmembrane interactions. This work was supported by the NIH Grants R01 NS088479 (L.P.W.), including a minority supplement (J.B.A.), and R35 GM118091 (H.-X.Z.).

## Author contributions

J.B.A., H.-X.Z., and L.P.W. designed research; J.B.A., A.G., and L.P.W. carried out and/or analyzed the functional experiments; X.L. and H.-X.Z., performed the computational

studies, including modeling and molecular dynamic simulations; J.B.A., X.L., H.-X. Z., and L.P.W. wrote the paper.

## Additional information

**Competing interests:** The authors declare no competing interests.

