## [Peer Review File · Nature Communications]

Reviewers' comments:

Reviewer #1 (Remarks to the Author):

This work examines the role of several disease associated mutations on the M4 helix of GluN1, GluN2A, or GluN2B subunits of the NMDA receptors. Through electrophysiological studies (whole-cell and single channel recordings), surface expression experiments, and molecular dynamics simulations, the authors have proposed a mechanism to describe the observed changes when a conserved glycine in the extreme extracellular third of M4 helix is mutated to an arginine (disease associated) or an alanine. Overall, the evidence provides support for the claims of the authors. There are a few minor issues which should be addressed, nevertheless.

Comments:

1: There is no chelating reagent in the external solution of whole cell recordings to chelate background zinc contamination for the GluN2A containing NMDAR constructs, while 50 μ M EDTA in the external solution for single channel recordings is used. This is with regards to data presented in Figure 2, Supplemental Table 2, and Supplemental Figure 2. The whole-cell recordings were carried out at pH 7.2, where zinc inhibition can be very significant (>50%). This zinc-inhibition desensitization is a classic presentation for background zinc-contamination, and accounts for a rather large measured desensitization for the GluN1/GluN2A receptors (~60%). Later in the manuscript, in Figure 5d, the authors show that when they incorporate EDTA into the recording solution, the extent of desensitization for GluN1/GluN2A receptors falls to ~30% from ~60% in Supplementary Figure 2C. Similar argument can be made about deactivation kinetics, where they are slower when inhibited by zinc (~81 ms) than when zinc has been chelated (44 ms). As background level of zinc-contamination can vary and is not well controlled, it would be more prudent to carry out the experiment in the presence of EDTA, or using buffered zinc concentrations.

2: It would be interesting to see how the open structure model, obtained via MD simulations in this study, compares with a recent NMDA receptor publication (Song et. al., 2018) where MD simulation was also used to simulate the open GluN1/GluN2A NMDA conformation.

Minor comments:

Page 7 paragraph 3, line 2: hGluN2(G826E) should have been hGluN2B(G826E)

The authors have reported the deactivation τ_{weighted} only in supplementary table 2. It would be useful to show both the extent and rates of fast and slow τ .

Reviewer #2 (Remarks to the Author):

In this paper Amin et al. have used a combination of single channel recordings, whole cell recordings, imaging, and MD simulations to study the role of glycine in the transmembrane segments of NMDA receptors. The reason for pursuing these investigations is the fact that there are several inherited missense mutations at these sites that are associated with neurological disorders. This study is thorough using both functional and computational methods to gain a deeper understanding of the role of the glycines. These studies show hotspots at specific glycine residues such, as 815 on GluN1, 819 in GluN2A, and 820 in N2B that are critical with even conservative substitutions such as alanine leading to severe effects. MD simulations provide the structural evidence pointing to the importance of this glycine in maintaining channel gating, with the alanine mutant disrupting the selectivity filter. Additionally the simulations show that the effect is observed as far as the CTD, showing differences in

splaying. Data on the triheteromeric receptors with the dead G820E mutant illustrates an interesting point that the receptor can be partly functional with three functional subunits.

Minor comments:

The text on pages 7 and 9 has Figures 2 a-h listed incorrectly.

While data presented show clear differences between GluN1-GluN2A and the mutant in Figure 3, the image shown does not clearly show the differences. A better representative image is required.

Reviewer #3 (Remarks to the Author):

NMDA receptors are centrally involved in almost every aspect of excitatory neurotransmission in the CNS and aberrant ion channel function in these receptors therefore often results in psychiatric or neurological disorders. This excellent and comprehensive study demonstrate that missense mutations located in the M4 segment of NMDA receptor subunits alter ion channel function and thereby provide a link to the associated clinical pathologies. Furthermore, by gleaning at the effects of these missense mutations, the authors reveal important new insight to the function of NMDA receptors. The study demonstrates that a conserved glycine in the M4 segment may act as a hinge that permits the intracellular portion of the ion channel to expand laterally, resulting in increased stability of long-lived open states. The longer dwell time in these long-lived open states slow deactivation and mediate high Ca²⁺ permeability of wild type NMDA receptors. These findings are novel and represent a lateral conceptual leap in our understanding of NMDA receptor structure-function. Furthermore, the authors suggest a movement of the M4 segment during gating, which provides a potential link to intracellular modulation of NMDA receptor function as well as metabotropic NMDA receptor signaling, which has gained considerable interest in recent years. The study therefore explores an important and timely topic and is of broad interest in the NMDA receptor field. Overall, the findings of this study can prove useful for future studies and will likely stimulate new experiments.

The manuscript is nicely written, the experiments are carefully constructed, the results are appropriately interpreted, and the figures are of excellent quality. The conclusions drawn are reasonable and substantiated. There are very few specific points that the authors should consider to improve the manuscript.

1) The suggestion that the M4 segments can serve as a structural substrate for NMDA receptor modulation is exciting, and this study is, to the best of my knowledge, the first that directly indicates motions of the M4 segment during NMDA receptor gating. This point is only briefly explored in the Discussion and the manuscript could benefit from a bit more elaboration on this topic. How might the findings that glycine binding, but not glutamate binding, primes NMDA receptors for internalization relate to this study? Are the authors envisioning that the splaying of M4 can happen in the absence of pore dilation?

2) In Supplemental Table 1, the GluN2A Leu812Thr mutation should be Leu812Met.

Manuscript: NCOMMS-18-11503-T (Amin et al.)

Response to Reviewers

We greatly appreciate the Reviewers for their thoughtful comments. As outlined below, we have incorporated all comments/suggestions/requests into the revised version of the manuscript in some form. We believe the manuscript is now a better read and more accurately reflects the data shown.

We present our point-by-point response in blue.

Reviewer #1 (Remarks to the Author):

Comments:

1: There is no chelating reagent in the external solution of whole cell recordings to chelate background zinc contamination for the GluN2A containing NMDAR constructs, while 50 μ M EDTA in the external solution for single channel recordings is used. This is with regards to data presented in Figure 2, Supplemental Table 2, and Supplemental Figure 2. The whole-cell recordings were carried out at pH 7.2, where zinc inhibition can be very significant (>50%). This zinc-inhibition desensitization is a classic presentation for background zinc-contamination, and accounts for a rather large measured desensitization for the GluN1/GluN2A receptors (~60%). Later in the manuscript, in Figure 5d, the authors show that when they incorporate EDTA into the recording solution, the extent of desensitization for GluN1/GluN2A receptors falls to ~30% from ~60% in Supplementary Figure 2C. Similar argument can be made about deactivation kinetics, where they are slower when inhibited by zinc (~81 ms) than when zinc has been chelated (44 ms). As background level of zinc-contamination can vary and is not well controlled, it would be more prudent to carry out the experiment in the presence of EDTA, or using buffered zinc concentrations.

This is an excellent point and certainly something we considered! However, our goal with the human missense mutations was not to get detailed mechanistic information but rather to get a general idea of how they affect function and how they might do so under 'physiological' conditions. We therefore recorded all human missense mutants in conditions that approximated physiological conditions (1 mM Ca^{2+} , pH 7.2 and with contaminating zinc) just to get a general idea of how they might affect function. Subsequent to identifying that many of these missense mutations affect receptor function (deactivation, Figure 2, and desensitization, Supplemental Figure 2), we then transitioned to more mechanistic questions and used a solution that had no added divalents, EDTA as well as at pH 8 to maximize single channel activity (single channel recordings, Figure 3). Hence, for all of our mechanistic questions concerning the conserved glycine, we used this solution (except for when measuring Ca^{2+} permeability). Finally, we should also note that for the human missense mutations every construct was recorded on at least 3-4 different transfection cycles, and correspondingly a variety of external solutions were used (with presumed variations in contaminating zinc). Hence, on average, variations in contaminating zinc would be averaged out.

In any case, to avoid any confusion and to make the reader aware of our goal and this transition in solutions (from 'physiological' to 'mechanistic' solutions) more clear, we have split apart original Figure 2, which had deactivation (physiological solution) and single channels (biophysical solution) in the same figure, into two figures. Figure 2 has just deactivation and Figure 3 the single channels. We also now explicitly state in the text the change in solutions and our rationale for doing so (pp. 6 & 7).

2: It would be interesting to see how the open structure model, obtained via MD simulations in this study, compares with a recent NMDA receptor publication (Song et. al., 2018) where MD simulation was also used to simulate the open GluN1/GluN2A NMDA conformation.

This is an extremely good suggestion. In Song et al. the open state was simulated by disrupting the helix bundle at in M3 with mutants known to disrupt the gate (GluN1 A650R and GluN2B A648R) in SYTANLAAF motif, much like the lurcher motif. Then they reversed the mutations and verified that the wild type remained in the open state. Unfortunately, they do not provide public access to this modeled open state, nor do they show anything in detail besides the pore lining structures (M2 & M3). We have compared the M2 loops in our open state 'model' and like Song et al., we find that they are staggered – the GluN1 N-site asparagine and the GluN2B N+1-site asparagine are at the same level as Song et al., also found (and which makes the senior author happy)!

Although we see this consistency with Song et al., we made no attempt to validate the open state model of wild type nor do we want it the focus of the present manuscript. To test the wild type model properly will take an extensive amount of effort. Rather, our approach was to compare 'relative' structures: we generated simulations for both gly-ala substitutions in GluN1 and GluN2 subunits and compared these models to that for wild type. This approach is not perfect but provides a 'general' framework for how local structures might be displaced in wild type (rather than an absolute model). Hence, in general we assume that the M4 segments permit displacement of the lower half of the ion channel but we do not know all the details, nor can we without a high resolution structure (cryo-EM or X-ray crystallography) of the NMDAR open state.

Minor comments:

Page 7 paragraph 3, line 2: hGluN2(G826E) should have been hGluN2B(G826E)

We thank the Reviewer for noticing this error and have corrected the mistake.

The authors have reported the deactivation $\tau_{weighted}$ only in supplementary table 2. It would be useful to show both the extent and rates of fast and slow τ .

We agree with the Reviewer and as requested we have added this information to Supplemental Table 2.

Reviewer #2 (Remarks to the Author):

Minor comments:

The text on pages 7 and 9 has Figures 2 a-h listed incorrectly.

We thank the Reviewer for noticing these errors and we have corrected the writing accordingly.

While data presented show clear differences between GluN1-GluN2A and the mutant in Figure 3, the image shown does not clearly show the differences. A better representative image is required.

We assume the Reviewer is talking about the inset images to Figure 3a. We agree that the images are not the best. Although we searched for new images, we were unable to find one that we felt showed a more distinct difference. Part of the problem is that even at pH 5.5, every

image has some background fluorescence. This background fluorescence presumably reflects receptors still residing in the endoplasmic reticulum, where the pH is around 7.2. We also prefer to show images that are close to the mean as the images now shown are. In any case, we are not crazed about this image either but could not find a better alternative.

Reviewer #3 (Remarks to the Author):

1) The suggestion that the M4 segments can serve as a structural substrate for NMDA receptor modulation is exciting, and this study is, to the best of my knowledge, the first that directly indicates motions of the M4 segment during NMDA receptor gating. This point is only briefly explored in the Discussion and the manuscript could benefit from a bit more elaboration on this topic. How might the findings that glycine binding, but not glutamate binding, primes NMDA receptors for internalization relate to this study? Are the authors envisioning that the splaying of M4 can happen in the absence of pore dilation?

The Reviewer raises some exciting questions and we too are very intrigued about how the M4 segments might contribute to functions besides modulating the core gating machinery. At present however we do not have any of these data, though we are starting to test these questions. Anyway, as requested we have altered the end of the Discussion to expand on some of these additional points including citing Nong et al., 2003, Nature (internalization) and Nabavi et al., 2013, PNAS (metabotropic actions).

2) In Supplemental Table 1, the GluN2A Leu812Thr mutation should be Leu812Met.

We thank the Reviewer for noticing this error and have corrected Supplemental Table 1 accordingly.

REVIEWERS' COMMENTS:

Reviewer #2 (Remarks to the Author):

The authors have addressed my concerns.

Reviewer #3 (Remarks to the Author):

The authors have addressed all the concerns and the manuscript is improved. This is high-quality work that provides important new insight to NMDA receptor structure and function.